# SIG: Self-Interpretable Graph Neural Network for Continuous-time Dynamic Graphs

## Abstract

While graph neural networks have demonstrated potential across various applications, explaining their predictions on dynamic graphs remains largely underexplored. This paper introduces a new research task: self-interpretable GNNs for continuous-time dynamic graphs (CTDGs). We aim to predict future links within dynamic graphs while simultaneously providing causal explanations for these predictions. There are two key challenges: (1) capturing the underlying structural and temporal information that remains consistent across both independent and identically distributed (IID) and out-of-distribution (OOD) data, and (2) efficiently generating high-quality link prediction results and explanations. To tackle these challenges, we propose a novel causal inference model, namely the Independent and Confounded Causal Model (ICCM). ICCM is then integrated into a deep learning architecture that considers both effectiveness and efficiency. Extensive experiments demonstrate that our proposed model significantly outperforms existing methods across link prediction accuracy, explanation quality, and robustness to OOD data. Our code and datasets are anonymously released at https://github.com/2024SIG/SIG.

## 1 Introduction

Graph neural networks (GNNs) have demonstrated remarkable efficacy in representing graph-structured data. However, the inherent opacity of GNNs poses challenges in comprehending and trusting their predictions, particularly in high-stakes domains such as fraud detection in financial systems (Kumar et al., 2019) or disease progression prediction in healthcare (Li et al., 2021), where interpretability is important.

Recent advancements in explainable Graph Neural Networks (GNNs) have aimed to unravel the underlying rationale guiding GNN predictions (Yuan et al., 2022). These models are broadly classified into two categories: *post-hoc interpretable models* (Ying et al., 2019; Xia et al., 2022; Luo et al., 2020; Lv & Chen, 2023) and *self-interpretable models* (Zhang et al., 2022a; Wu et al., 2022b; Liu et al., 2022a). Post-hoc interpretable models focus on elucidating the behaviors of the primary predictive GNN model after its construction without altering its structural or training aspects. Conversely, self-interpretable models are inherently transparent in their decision-making processes, obviating the requirement for additional post-hoc techniques. Existing self-interpretable models include decision trees (Kotsiantis, 2013), subgraph extraction based models (Liu et al., 2024; Yin et al., 2024; Deng & Shen, 2024), attention-based mechanisms (Shu et al., 2019), rule-based models (Setnes et al., 1998; Geng et al., 2023), and causal inference models (Wu et al., 2022b; Zhang et al., 2022b). These models inherently provide interpretability, offering explanations of their predictions without relying on auxiliary interpretive methods.

This work tackles a novel research task: *the development of self-interpretable models for continuous-time dynamic graphs (CTDGs)*. Unlike static graphs or discrete-time dynamic graphs (DTDGs), CTDGs continuously evolve with time, enabling more precise modeling of dynamic processes. However, achieving interpretability in CTDGs presents two challenges. **The first challenge is susceptibility to shortcut features**, which is a prevalent issue in most existing self-interpretable models. Shortcut features are patterns that provide good performance on test data but fail to generalize to out-of-distribution (OOD) data (Geirhos et al., 2020). Recent causal inference methods (Wu et al., 2022b; Fan et al., 2022; Sui et al., 2022; Liu et al., 2022a) have been developed to address this

challenge by performing interventions on graphs. However, they are based on static graphs or DT-DGs, and cannot effectively handle the CTDGs. **The second challenge is the efficiency of the self-interpretable model**. This challenge is amplified in CTDGs due to their constantly evolving structures. Different from static or discrete-time graphs, CTDGs undergo continuous node and edge additions/deletions, resulting in a much larger number of possible topologies. This significantly increases the computational burden of performing interventions in causal inference models for CTDGs. Exhaustive sampling of topologies becomes computationally expensive, while limited sampling might hinder model effectiveness.

Designing self-interpretable models for CTDGs is intricate, as the model must meet three critical requirements: (1) Handle both independent and identically distributed (IID) and out-of-distribution (OOD) data; (2) Capture invariant subgraphs in both structural and temporal aspects; (3) Perform interventions efficiently. To fulfill these requirements, we propose a novel *self-interpretable GNN (SIG)* method. SIG initiates with a theoretical analysis of the problem from a causal effect perspective and proposes a novel causal inference model, namely the Independent and Confounded Causal Model (ICCM). ICCM incorporates two key components: the *Independent Causal Model* (ICM) and the *Confounded Causal Model* (CCM). The ICM is designed for IID data, where the causal subgraph is the unique exogenous variable influencing the predictive label. In contrast, the CCM is tailored for OOD data, where shortcut features act as confounding factors, creating backdoor paths that result in spurious correlations between causal subgraphs and prediction labels. SIG employs interventions to disrupt these "backdoor paths" and mitigate the influence of confounding factors in CCM. To achieve efficient intervention optimization, SIG utilizes the Normalized Weighted Geometric Mean (NWGM) (Xu et al., 2015) instead of directly pairing causal subgraphs or their representations with each element in the confounders set. During implementation, SIG leverages a deep learning clustering technique to approximate the actual confounders within CTDGs, thereby reducing computational costs. SIG generates final predictions based on both temporal and structural representations from the CTDG, along with these confounders.

The main contributions of this paper are summarized as follows:

- We investigate a new research task on CTDGs, which outputs not only the prediction label but also a concise causal subgraph for the prediction. To the best of our knowledge, the proposed SIG is the first self-interpretable GNN for CTDGs that is capable of handling both IID and OOD data.

- We present a thorough causal analysis of SIG, elucidating the causal effects and underlying mechanisms. This theoretical analysis serves as the foundation for our innovative model design and optimization strategies.

- We develop a novel deep learning framework that implements the theoretically established causal models, effectively and efficiently addressing challenges of self-interpretability on CTDGs.

- Extensive experiments on five real-world datasets demonstrate that SIG significantly outperforms state-of-the-art methods in link prediction, graph explanation, and handling OOD datasets.

## 2 RELATED WORK

**Dynamic Graph Neural Networks.** Dynamic graph neural networks encompass two primary classifications: Discrete-Time Dynamic Graphs (DTDGs) (Ma et al., 2019; Qin et al., 2023) and Continuous-Time Dynamic Graphs (CTDGs) (Li et al., 2017; De Winter et al., 2018; Li et al., 2023). DTDGs comprise a sequence of static graph snapshots captured at regular time intervals (Sankar et al., 2020; Yu et al., 2018; Mahdavi et al., 2018). CTDGs capture the evolution of graphs by considering modifications on the graph that occur continuously rather than discretely at predefined time steps (Xu et al., 2020; Trivedi et al., 2019; Feng et al., 2023). However, these GNNs focus on modeling graph dynamics and fail to offer sufficient interpretability for the underlying processes.

**Explainability of Graph Neural Networks.** The majority of existing explainable GNNs fall into the category of post-hoc interpretable GNNs. These frameworks are formulated as an optimization task that maximizes the mutual information between a GNN's prediction and distribution of possible subgraph structures (Ying et al., 2019; Luo et al., 2020; Xia et al., 2022; Rossi et al., 2022). However, post-explanation methodologies may encounter inaccuracies or incompleteness in elucidating the genuine reasoning process of the underlying model and require iterative executions of

the prediction model to delve into the intricate relationships between inputs and outputs, consequently incurring a notable computational overhead. Few efforts are devoted to self-interpretable GNNs. Prototype-based methods (Zhang et al., 2022a) learn prototype vectors as explanations. These methods either fail to produce an explainable subgraph or depend on computationally expensive subgraph exploration techniques. Neighborhood-based methods (Dai & Wang, 2021), while effective in capturing local node and structure similarity, might include many extraneous details or miss out on key structural patterns that are critical for interpretation. They often fail to distill the graph into a form that clearly communicates the rationale behind a model's prediction. Subgraph extraction-based methods (Liu et al., 2024; Zheng et al., 2023; Yin et al., 2024; Deng & Shen, 2024; Feng et al., 2022; Liu et al., 2022b) identify the most influential subgraph for decision-making. They may neglect the influence of confounding factors, potentially leading to inaccurate explanations.

**Causal Inference on Graph Neural Networks.** Causal inference seeks to unveil and comprehend the causal variables responsible for observed phenomena. On real-world graphs, uncovering these causal variables becomes an act of explanation, revealing the "why" behind intricate relationships. Most existing methods focus on static graphs. These approaches either manipulate non-causal elements within a graph to create counterfactual graph data, as demonstrated in (Wu et al., 2022b), or utilize implicit interventions at the representation level, as shown in (Sui et al., 2022; Fan et al., 2022; Miao et al., 2022). The method most closely associated with this context is DIDA (Zhang et al., 2022b), an invariant rational discovery approach specifically designed for DTDGs. DIDA requires the construction of an intervention set for each node and snapshot. When the graph is divided into too many snapshots, applying DIDA becomes time-consuming. Conversely, dividing the graph into too few snapshots leads to a loss of significant time-related information. The correlations and differences between related studies and this work can be found in App. D.2

## 3 PROBLEM DEFINITION

This paper investigates a new problem of developing a self-interpretable graph learning model tailored for the analysis of continuous-time dynamic graphs, with a particular emphasis on its inherent capabilities for link prediction and explainability.

**Definition 1** (Continuous-Time Dynamic Graph (CTDG)). *A continuous-time dynamic graph $G = (\mathcal{V}, \mathcal{E}, \mathcal{T})$ comprises a set of vertices $\mathcal{V}$, a set of edges $\mathcal{E}$, and a time domain $\mathcal{T}$. This graph evolves continuously over time $t \in \mathcal{T}$, where at each time instance $t$, edges might undergo additions, removals, or changes in their characteristics. Formally, the graph $\mathcal{G}$ can be denoted as a sequence of edges $G = \langle e_{ij}(t_k) \rangle$. Each edge $e_{ij}(t_k)$ signifies an interaction occurring between the source node $v_i$ and the target node $v_j$ at time $t_k$. Additionally, we introduce $x_{ij}^e(t_k)$ to denote the feature vector of edge $e_{ij}(t_k)$, while $x_i^n$ indicates the feature vector of node $v_i$,*

**Definition 2** (Self-interpretable GNN for CTDG). *Given a CTDG $G$ and two distinct nodes, $v_i \in \mathcal{V}$ and $v_j \in \mathcal{V}$, the primary objectives of self-interpretable GNN are twofold: firstly, to accurately predict whether an edge will form between nodes $v_i$ and $v_j$; and secondly, to discern a causal subgraph that elucidates the underlying reasons for the prediction.*

## 4 CASUAL EFFECT LOOK

### 4.1 INDEPENDENT CAUSAL MODEL (ICM)

The link prediction label on dynamic graphs can be influenced by both the structural topology and temporal dynamics (Cong et al., 2023). Consequently, this paper proposes to capture causal information emanating from both the structural and the temporal perspectives, as shown in Figure 1 (a). In this subsection, we formalize the causal inference (Pearl et al., 2000) by inspecting the causalities among six variables: the input graph $G$, the structural causal subgraph $G_s$, the temporal causal subgraph $G_t$, the temporal feature $M^T$, the structural feature $M^S$ and the prediction label $Y^I$. The following equations summarize the core assumptions:

**Assumption 1** (ICM).

$$G_t, G_s := f_{ext}(G), \quad M^T := f_t^I(G_t), \quad M^S := f_s^I(G_s), \quad Y^I := f_o^I(M^T, M^S) \quad (1)$$

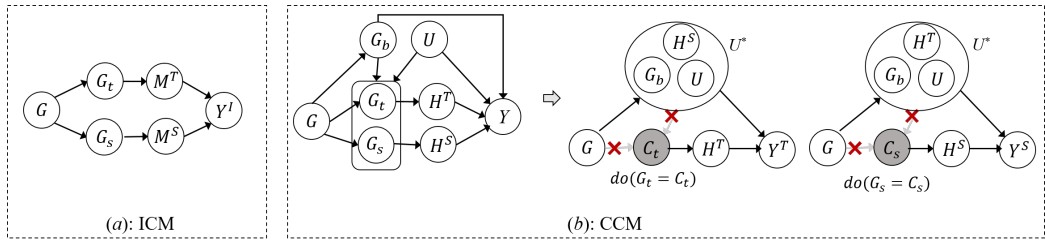

Figure 1: Causal Models.

In this assumption, $f_{ext}$ performs the extraction of causal subgraphs from the input graph $G$, $f_t^I(\cdot)$ and $f_s^I(\cdot)$ encode the causal subgraph into latent representations $M^T$ and $M^S$, respectively. $f_o^I$ calculates the ultimate prediction outcome using the $M^T$ and $M^S$.

## 4.2 CONFOUNDED CAUSAL MODEL (CCM)

To handle the confounding variables that may introduce bias in OOD data predictions, we introduce CCM. As shown in Figure 1 (b), CCM considers confounders consisting of non-causal subgraph $G_b$ and unobserved variables $U$, where $G_b$ is the residual part of the graph once the causal subgraphs are excluded. These confounders contain information about possible shortcut features, which could lead to spurious correlations between the causal subgraph and the prediction labels.

Let $G_*$ denote a causal subgraph which is either $G_s$ or $G_t$. To block the backdoor paths $G_* \leftarrow G_b \rightarrow Y$ and $G_* \leftarrow U \rightarrow Y$, we perform interventions on $G_*$. Specifically, we perform interventions as $do(G_s = C_s)$ based on structural features and $do(G_t = C_t)$ based on temporal features, where $C_s$ and $C_t$ are constant subgraphs. Through the replacement of $G_s$ with $C_s$ and $G_t$ with $C_t$, these interventions effectively block the backdoor paths, thereby eliminating the previously existing spurious correlation between $G_*$ and $Y$. The foundational assumptions guiding these models are summarized as follows:

**Assumption 2 (CCM).**

$$H^S := f_s^C(C_s), \quad H^T := f_t^C(C_t), \quad Y^S := f_o^S(H^S, U^*), \quad Y^T := f_o^T(H^T, U^*) \tag{2}$$

Within these formulations, $U^*$ denotes the set of confounders, which can be either $\{H^S, G_b, U\}$ or $\{H^T, G_b, U\}$. $f_s^C(\cdot)$ and $f_t^C(\cdot)$ are structural and temporal encoders, $f_o^S(\cdot)$ and $f_o^T(\cdot)$ are structural and temporal predictors, $Y^S$ and $Y^T$ represent the prediction labels resulting from structural and temporal interventions, respectively.

## 4.3 COMBINATION OF ICM AND CCM (ICCM)

This subsection discusses the Independent and Confounded Causal Model (ICCM), which serves as the foundation of our proposed SIG framework.

Recall that in ICM, we use $M^S$ and $M^T$ to capture the structural and temporal features from the causal subgraph. In CCM, we use $H^S$ and $H^T$ to denote the structural and temporal representations from the causal subgraph. To ensure consistency between these models, we define:

$$H^S := M^S, \quad H^T := M^T. \tag{3}$$

Assumption 3 summarizes the core principles guiding ICCM:

**Assumption 3 (ICCM).**

$$Y^I := f_o^I(H^S, H^T), \quad Y^S := f_o^S(H^S, U^*), \quad Y^T := f_o^T(H^T, U^*) \tag{4}$$

where $H^S$ and $H^T$ are structural and temporal representations from the causal subgraphs, $f_o^I$, $f_o^S$ and $f_o^H$ are linear networks followed by a sigmoid activation functions. The following equations present the mathematical formulation of ICCM:

$$P(Y^I = y^I | G) = \sigma\left(W_1^I f_y^{I_S}(H^S) + W_2^I f_y^{I_T}(H^T)\right), \tag{5}$$

Figure 2: The deep learning implementation of SIG.

$$P\big(Y^S = y^S | do(G_s = C_s)\big) = \mathbb{E}_{d \sim \mathcal{D}}[\sigma\big(W_1^c f_y^s(H^S) + W_3^c f_y^u(d)\big)], \qquad (6)$$

$$P\big(Y^T = y^T | do(G_t = C_t)\big) = \mathbb{E}_{d \sim \mathcal{D}}[\sigma\big(W_2^c f_y^t(H^T) + W_4^c f_y^u(d)\big)]. \qquad (7)$$

where $\mathcal{D}$ denotes the set of confounding factors, $W_*^*$ denotes the model parameters, $f_y^*(\cdot)$ denotes linear network, $\sigma$ denotes the activation function. Detailed explanations of Equation 6 and Equation 7 are in App. C.

Equations 6 and 7 require evaluating the model for each confounder $d \in \mathcal{D}$ with both $H^S$ and $H^T$. This becomes computationally expensive for large temporal networks. To address this, we leverage the Normalized Weighted Geometric Mean (NWGM) approximation (Xu et al., 2015), i.e., $\mathbb{E}_{d \sim \mathcal{D}}[\sigma\big(W_*^c f_y^s(H^S) + W_*^c f_y^u(d)\big)] \approx \sigma\big(\mathbb{E}_{d \sim \mathcal{D}}[W_*^c f_y^s(H^S) + W_*^c f_y^u(d)]\big)$. After applying NWGM, Equations 6 and 7 can be reformulated as:

$$P\big(Y^S = y^S | do(G_s = C_s)\big) \approx \sigma\big(W_1^c f_y^s(H^S) + W_3^c \mathbb{E}_{d \sim \mathcal{D}}[f_y^u(d)]\big), \qquad (8)$$

$$P\big(Y^T = y^T | do(G_t = C_t)\big) \approx \sigma\big(W_2^c f_y^t(H^T) + W_4^c \mathbb{E}_{d \sim \mathcal{D}}[f_y^u(d)]\big). \qquad (9)$$

The designed causal model ICCM is exploited as the theoretical underpinning for implementing our deep learning framework, which will be presented in the next section.

## 5 DEEP LEARNING IMPLEMENTATION

### 5.1 OVERVIEW

ICCM relies on structural and temporal representations derived from constant causal subgraphs $C_s$ and $C_t$. However, in real-world scenarios, these causal subgraphs are typically unobserved. To address this issue, SIG employs two causal subgraph extractors to extract structural and temporal subgraphs $\hat{C}_s$ and $\hat{C}_t$ from the input data. These extracted subgraphs are then used to approximate $C_s$ and $C_t$. Figure 2 illustrates the overall structure of the SIG framework. First, the causal subgraph extraction aims to identify structural and temporal subgraphs $\hat{C}_s$ and $\hat{C}_t$. These subgraphs are then encoded into hidden representations $H^S$ and $H^T$. Subsequently, the confounder generation component produces a confounder dictionary $\hat{\mathcal{D}}$. Finally, both $H^S$ and $H^T$ are passed to the classifier $f_o^I$ to generate $y^I$. Simultaneously, along with the produced confounder dictionary, $H^S$ and $H^T$ are also fed into classifiers $f_o^S$ and $f_o^T$ to output $y^S$ and $y^T$, respectively. We will delve deeper into the details of each module in the following sections.

### 5.2 CAUSAL SUBGRAPH EXTRACTING AND ENCODING

**Temporal causal subgraph extraction and encoding.** Given a dynamic graph $G$ and two nodes ($u$ and $v$) for prediction, we initially generate two edge sequences $S_u$ and $S_v$ by selecting the top $N$ most recent temporal edges linked to $u$ and $v$, respectively. The parameter $N$ functions as a dataset-specific hyper-parameter. If the number of edges linked to a node is fewer than $N$, all available connections will be retained. For each edge $e_{ui}(t_k) \in S_u$, a temporal encoding is performed using

$\cos\big((t_0 - t_k)\omega\big)$ (Cong et al., 2023), where $t_0$ denotes the timestamp used for predicting the edge's existence, $\omega = \big\{\alpha^{-(i-1)/\beta}\big\}_{i=1}^{d}$, with $\alpha$ and $\beta$ representing hyperparameters. This encoding is combined with its corresponding edge features as $[\cos\big((t_0 - t_k)\omega\big)\|\mathbf{x}_{ui}^e(t_k)]$.

Let $F_u^{(0)}$ denote the stack of edge features within the sequence $S_u$. A 1-layer MLP-mixer (Tolstikhin et al., 2021) is employed to produce the final temporal representations, i.e., $F_u = \text{MLP-mixer}(F_u^{(0)})$. Two queries and keys are generated for node $u$ and $v$ using: $q_u = W_1^m \text{Mean}(F_u)$, $K_v = W_2^m(F_v)$, $q_v = W_1^m \text{Mean}(F_v)$, $K_u = W_2^m(F_u)$. The subgraph is generated by:

$$M_v^e = \text{Softmax}\left(\frac{q_u^T K_v}{\sqrt{d}}\right), \quad M_u^e = \text{Softmax}\left(\frac{q_v^T K_u}{\sqrt{d}}\right), \quad \hat{C}_t = \text{TOP}_k(M_u^e, M_v^e). \quad (10)$$

Here, $d$ denotes a specific hyperparameter, and $M_*^e[k]$ represents the importance score assigned to the $k$-th edge within $S_*$. Consequently, the highest top-$k$ scores in $M_v^e$ and $M_u^e$ are selected to construct the temporal causal subgraphs $\hat{C}_t$. Finally, the temporal representation $H^T$ is encoded by:

$$h_u^t = \text{Mean}\big(\{F_v | v \in \mathcal{N}_T(u)\}\big), \quad h_v^t = \text{Mean}\big(\{F_u | u \in \mathcal{N}_T(v)\}\big), \quad H^T = [h_u^t \| h_v^t], \quad (11)$$

where $\mathcal{N}_T(u)$ denotes the nodes linked to $u$ in $\hat{C}_t$.

**Structural causal subgraph extraction and encoding.** Structural node representation is encoded based on its n-hop neighborhood: $z_u = x_u^n + \text{Mean}\big(\{x_v^n \mid v \in \mathcal{N}^n(u; t_0 - T, t_0)\}\big)$. Here, $\mathcal{N}^n(u; t_0 - T, t_0)$ denotes the n-hop neighbors of node $u$ with edge timestamps ranging from $t_0 - T$ to $t_0$, where $T$ represents a dataset-specific hyperparameter. The node mask matrices are computed through the equations:

$$M_v^n = \text{Softmax}\left(\frac{z_u^T Z_v}{\sqrt{d}}\right), \quad M_u^n = \text{Softmax}\left(\frac{z_v^T Z_u}{\sqrt{d}}\right), \quad \hat{C}_s = \text{TOP}_k(M_u^n, M_v^n). \quad (12)$$

Here, $Z_u$ and $Z_v$ is the stack of the encoded node features of all nodes in $\mathcal{N}^n(u; t_0 - T, t_0)$ and $\mathcal{N}^n(v; t_0 - T, t_0)$, respectively. The nodes with the highest top-$k$ scores in $M_u^n$ and $M_v^n$ are chosen to form the structural causal subgraph. The final structural representation $H^S$ is computed by:

$$h_u^s = x_u^n + \text{Mean}\big(\{x_i^n | i \in \mathcal{N}_S^n(u)\}\big), \; h_v^s = x_v^n + \text{Mean}\big(\{x_i^n | i \in \mathcal{N}_S^n(v)\}\big), \; H^S = [h_u^s \| h_v^s], \; (13)$$

where $x_v^n$ represents the node feature of $v$, $\mathcal{N}_S^n(u)$ represents the n-hop neighbors of $u$ in $\hat{C}_s$.

## 5.3 CONFOUNDERS GENERATION

The confounder dictionary is expected to contain representations of all confounders, including non-causal subgraphs and unobserved factors. In this paper, we collect the representation of each link's temporal and structural subgraph, which includes both the causal subgraph and the confounder for that link. It is important to note that the causal subgraph for one link may serve as the non-causal subgraph for another. Therefore, it is reasonable to use these representations to approximate potential confounders. As the dynamic graph evolves, the number of distinct confounders becomes extremely large, making it computationally expensive to account for all of them. To address this, we cluster the representations and use each cluster centroid to approximate a group of similar confounders.

Specifically, given the dynamic graph $G = (\mathcal{V}, \mathcal{E}, \mathcal{T})$, we adopt a dynamic GNN encoder (Cong et al., 2023) to extract the representations for each link in $\mathcal{E}$ based on its temporal and structural subgraph, resulting in the matrix $\mathcal{X} \in \mathbb{R}^{|\mathcal{E}| \times l}$, where $l$ denotes the embedding dimension. By utilizing the deep learning clustering method VaDE (Jiang et al., 2016), we group $\mathcal{X}$ into $k$ clusters, i.e., $\{C_1, \ldots, C_k\} = \text{VaDE}(\mathcal{X})$. The centroids within each cluster serve as indicators of the central tendencies, effectively summarizing the overall features or characteristics among subgraph information within the same cluster. Consequently, computing the cluster-wise average yields a representation for each cluster, resulting in a confounder dictionary with the shape $\hat{\mathcal{D}} \in \mathbb{R}^{k \times l}$, where $\hat{\mathcal{D}}[i] = \text{Mean}(C_i)$. Finally, the expectation of confounders is computed by:

$$\mathbb{E}_{d \sim \hat{\mathcal{D}}}[f_y^u(d)] = \sum_{i=1}^{|\hat{\mathcal{D}}|} \alpha_i \hat{\mathcal{D}}[i], \quad [\alpha_1, \alpha_2, \ldots, \alpha_k] = \text{Softmax}\left(\frac{(W_1^c \hat{\mathcal{D}})^T W_2^c q}{\sqrt{|q|}}\right) \quad (14)$$

where $W_1^c$ and $W_2^c$ are learnable matrices. We set $q = H^S$ and $q = H^T$ for $y^S$ and $y^T$, respectively.

## 5.4 Prediction and Optimization

Given representations $H^S$ and $H^T$, and the expectation of confounders $\mathbb{E}_{d \sim \hat{\mathcal{D}}}[f_y^u(d)]$, we can make the final predictions based on Equations 5, 8, and 9.

Intuitively, if a subgraph $\hat{C}_*$ is irrelevant to the final prediction $Y$, then changing the subgraph should not affect the prediction. In other words, a subgraph that is relevant to the prediction should have high mutual information with the label. Formally, the learning objectives of the proposed model can be formulated as follows:

$$\max_{\Omega} I(\hat{C}_s, Y) + I(\hat{C}_t, Y), s.t. \ \hat{C}_s \perp U^*, \hat{C}_t \perp U^* \tag{15}$$

where $\Omega$ is the set of model parameters. $I(\hat{C}_*, Y)$ is the mutual information between the causal subgraph $\hat{C}_*$ and the label $Y$, $\hat{C}_* \perp U^*$ means that $\hat{C}_*$ is independent of the unobserved variables $U^*$.

Maximizing mutual information is equivalent to minimizing a variational upper bound of the risk functions (Alemi et al., 2016; Yu et al., 2020). Hence, we define the total learning objective of SIG as :

$$\mathcal{L} = \lambda_i \mathcal{R}_i(y^I, y) + \lambda_t \mathcal{R}_t(y^T, y) + \lambda_s \mathcal{R}_s(y^S, y), \tag{16}$$

where $\mathcal{R}_i, \mathcal{R}_t, \mathcal{R}_s$ are risk functions of IID prediction, temporal intervention prediction, and structural intervention prediction, respectively. $\lambda_*$ are hyperparameters, and $y$ is ground-truth label. This paper adopts cross-entropy loss as risk functions $\mathcal{R}_*$. Details are in App. C.2.

## 6 Experiments

In this section, we conduct extensive experiments on five dynamic graph datasets. Our experiments aim to answer the following questions:

- **RQ1:** Does SIG improve the performance of methods for link prediction in dynamic graphs?
- **RQ2:** What is the explanation capability of SIG?
- **RQ3:** How well does SIG perform in mitigating OOD issues?

### 6.1 Experimental Settings

**Dataset and evaluation metrics.** We conduct experiments on five real-world datasets: Wikipedia, Reddit, MOOC, LastFM and SX. We employ the average precision (AP) and area under the curve (AUC) as the evaluation metrics for link prediction. We adopt fidelity (FID) w.r.t. sparsity (SP) as the evaluation metrics for graph explanation. Details are in App. E.1 and E.2.

**Baselines.** Note that the proposed SIG is the first self-interpretable GNN specifically designed for CTDGs. Given the limited studies in self-interpretable GNNs for dynamic graphs, our evaluation spans several comparisons by considering different types of baselines. (1) Initially, SIG undergoes comparison with *three existing dynamic GNN models*: TGN (Rossi et al., 2020), TGAT (Xu et al., 2020), GM_ori and GM_50n (Cong et al., 2023). These models are designed to handle CTDGs.

Table 1: Comparison with SOTA graph link prediction models w.r.t. AUC and AP. The best scores are highlighted in **bold**, and the second highest scores are highlighted in underline.

| Model | Wikipedia | | Reddit | | MOOC | | LastFM | | SX | |
|---|---|---|---|---|---|---|---|---|---|---|
| | AP | AUC | AP | AUC | AP | AUC | AP | AUC | AP | AUC |
| TGN | 95.54 | 95.06 | 95.96 | 96.16 | 79.56 | 81.73 | 79.03 | 77.90 | 68.28 | 73.64 |
| TGAT | 97.25 | 96.92 | 98.20 | 98.12 | 86.91 | 88.44 | 82.46 | 80.97 | 71.44 | 74.01 |
| GM_ori | 99.75 | 99.79 | 99.90 | 99.91 | 99.91 | 99.93 | 96.16 | 97.73 | 97.60 | 97.62 |
| GM_50n | 99.69 | 99.73 | 99.92 | 99.93 | 99.83 | 99.86 | 96.18 | 97.49 | 96.94 | 96.97 |
| DIDA | 86.46 | 89.09 | 83.04 | 81.72 | 97.47 | 98.43 | 55.56 | 54.57 | 92.33 | 91.42 |
| SIG | **99.94** | **99.94** | **99.99** | **99.99** | **99.95** | **99.97** | **99.96** | **99.98** | **99.71** | **99.70** |

Table 2: Comparison with SOTA explanation models. 'TLE' indicates that the time limit of 24 hours was exceeded. 'FID(SP)' denotes the best fidelity value FID along with its corresponding occurred sparsity SP (SP $\in \{0.2, 0.4, 0.6, 0.8, 1.0\}$). 'AUFSC' stands for the Area Under the Fidelity-Sparse Curve.

| Type | | Model | Wikipedia | | Reddit | | MOOC | | LastFM | | SX | |
|------|---|-------|-----------|-------|--------|-------|------|-------|--------|-------|------|-------|
| | | | FID(SP) | AUFSC | FID(SP) | AUFSC | FID(SP) | AUFSC | FID(SP) | AUFSC | FID(SP) | AUFSC |
| Post-hoc | TGAT | ATTN | 18.92(1.0) | 3.36 | TLE | TLE | TLE | TLE | TLE | TLE | TLE | TLE |
| | | PBONE | 18.92(1.0) | 2.57 | TLE | TLE | TLE | TLE | TLE | TLE | TLE | TLE |
| | | PGExp | 18.92(1.0) | 3.18 | TLE | TLE | TLE | TLE | TLE | TLE | TLE | TLE |
| | | TGExp | TLE | TLE | TLE | TLE | TLE | TLE | TLE | TLE | TLE | TLE |
| | TGN | ATTN | 23.90(1.0) | 9.48 | TLE | TLE | TLE | TLE | TLE | TLE | TLE | TLE |
| | | PBONE | 23.90(1.0) | 7.73 | TLE | TLE | TLE | TLE | TLE | TLE | TLE | TLE |
| | | PGExp | 23.90(1.0) | 7.92 | TLE | TLE | TLE | TLE | TLE | TLE | TLE | TLE |
| | | TGExp | TLE | TLE | TLE | TLE | TLE | TLE | TLE | TLE | TLE | TLE |
| Self-int | | DIDA | 1.31(1.0) | 0.34 | 0(0) | -0.75 | 0(0) | -0.17 | 0(0) | -0.17 | 0(0) | -0.47 |
| | | SIG | **53.70(0.6)** | **42.09** | **58.21(0.4)** | **38.29** | **30.71(1.0)** | **6.47** | **28.29(0.2)** | **17.10** | **53.94(0.2)** | **25.86** |

Table 3: Comparison with SOTA graph explanation models on the sampled datasets.

| Type | | Model | Wikipedia_sample | | Reddit_sample | | MOOC_sample | | LastFM_sample | | SX_sample | |
|------|---|-------|------------------|-------|---------------|-------|-------------|-------|---------------|-------|-----------|-------|
| | | | FID(SP) | AUFSC | FID(SP) | AUFSC | FID(SP) | AUFSC | FID(SP) | AUFSC | FID(SP) | AUFSC |
| Post-hoc | TGAT | ATTN | 40.40(1.0) | 11.45 | 36.00(1.0) | 5.48 | 6.29(1.0) | 1.35 | 21.18(1.0) | 6.59 | 22.63(0.4) | 20.17 |
| | | PBONE | 40.40(1.0) | 6.88 | 36.00(1.0) | 6.51 | 6.29(1.0) | 0.88 | 21.18(1.0) | 6.35 | 22.33(0.2) | 20.97 |
| | | PGExp | 40.40(1.0) | 7.63 | 36.00(1.0) | 6.17 | 6.29(1.0) | 1.43 | 21.18(1.0) | 5.49 | 18.26(1.0) | 9.21 |
| | | TGExp | TLE | TLE | TLE | TLE | TLE | TLE | TLE | TLE | TLE | TLE |
| | TGN | ATTN | 27.62(0.8) | 14.66 | 20.43(1.0) | 5.49 | 1.12(1.0) | 0.63 | 2.79(1.0) | 1.13 | 1.82(0.8) | 0.95 |
| | | PBONE | 25.10(1.0) | 11.02 | 20.43(1.0) | 8.70 | 1.12(1.0) | 0.63 | 2.77(1.0) | 1.27 | 4.85(0.2) | 2.41 |
| | | PGExp | 25.10(1.0) | 10.84 | 20.43(1.0) | 3.03 | 1.23(0.8) | 0.69 | 2.79(1.0) | 1.24 | 1.04(0.4) | -0.44 |
| | | TGExp | TLE | TLE | TLE | TLE | TLE | TLE | TLE | TLE | TLE | TLE |
| Self-int | | DIDA | 0.90(0.6) | 0.34 | 0(0) | -0.75 | 0(0) | -0.07 | 0(0) | -0.17 | 0(0) | -0.47 |
| | | SIG | **54.58(0.6)** | **42.43** | **58.27(0.4)** | **38.19** | **17.82(0.2)** | **5.11** | **28.68(0.2)** | **17.88** | **52.63(0.2)** | **25.26** |

However, as they lack the capacity to produce explainable outcomes, our comparison primarily focuses on link prediction tasks across original datasets and synthetic OOD datasets. (2) Additionally, we compare the proposed model with *four post-interpretable models*, including an attention-based explainer (ATTN (Xia et al., 2022)), a perturbing-based explainer (PBONE (Xia et al., 2022)), a static graph explainer (PGExp (Luo et al., 2020)), and a dynamic graph explainer (TGExp (Xia et al., 2022)). These models were thoughtfully chosen to represent diverse graph explanation approaches. Given their post-interpretable nature, our comparison focuses solely on graph explanation tasks. (3) Further, we compare SIG with DIDA (Zhang et al., 2022b), *a self-interpretable GNN for DTDG*. Our comparative analysis with DIDA spans across all tasks. Details are in App. E.4.

## 6.2 COMPARISON WITH SOTA DYNAMIC GNNS (RQ1)

Table 1 illustrates a comparative analysis between SIG and recent dynamic graph neural networks w.r.t. link prediction tasks. Among all dynamic graph neural networks, GM_ori and GM_50n achieve higher AP and AUC values compared to TGAT and TGN. These results suggest that recurrent neural networks and self-attention mechanisms are not always essential for effective temporal graph learning.

DIDA, a self-interpretable GNN explicitly tailored for DTDGs, obtains low scores across multiple datasets. This disparity in performance stems from the finer granularity in modeling temporal dynamics offered by CTDGs compared to DTDGs. CTDGs enable a more precise representation of event occurrences, making them inherently more challenging.

SIG consistently outperforms all baselines across all datasets. These results highlight the effectiveness of SIG's novel causal inference model and its ability to capture complex temporal relationships within dynamic graphs, effectively removing the shortcut features that hinder performance.

## 6.3 COMPARISON WITH SOTA GRAPH EXPLANATION MODELS (RQ2)

Table 2 presents a comparative analysis between SIG and state-of-the-art graph explanation methodologies. In this comparison, the category of 'Post-hoc' block denotes the application of post-hoc interpretable models. Building upon prior techniques (Xia et al., 2022), we apply these post-hoc in-

terpretable models to two dynamic GNN models: TGAT and TGN. Meanwhile, the 'Self-int' block refers to the self-interpretable GNNs designed specifically for dynamic graphs.

Our empirical investigation reveals that all post-hoc interpretable models require over 24 hours to process the `Reddit`, `MOOC`, `LastFM`, and `SX` datasets. This extensive computational time is primarily attributed to their reliance on complex computation methodologies for extracting explainable subgraphs. For instance, TGExp utilizes Monte Carlo Tree Search for subgraph extraction, rendering it impractical when generating explanations for each prediction. Although DIDA manages to produce results within 24 hours, its explanatory performance significantly lags behind SIG. This occurs because, when transitioning from a cntinuous time dynamic graph to a discrete time dynamic graph, a significant amount of dynamic information is lost.

Given the prevalent occurrence of TLE issues in most models documented in Table 2, we sought to assess the efficacy of SIG against established baselines. To this end, we randomly sampled 500 edges from the datasets, following the methodology outlined in (Xia et al., 2022), thereby creating a test set of edges, as depicted in Table 3. Empirical results reveal that, on average, SIG outperforms the best baselines by 17.10% and 16.77% concerning FID(SP) and AUFSC, respectively. Notably, our observations indicate that SIG achieves best fidelity, particularly at sparsity levels below 0.6 across most datasets. Conversely, the majority of existing explainable methods attain best fidelity at a sparsity of 1. These outcomes underscore SIG's capability to discern the most distinctive subgraph as the explanation.

Table 4: Comparison on OOD datasets.

| Model | Reddit_OOD | | | | | | LastFM_OOD | | | | | | SX_OOD | | | | | |
|---|---|---|---|---|---|---|---|---|---|---|---|---|---|---|---|---|---|---|
| Split | 0.4 | | 0.6 | | 0.8 | | 0.4 | | 0.6 | | 0.8 | | 0.4 | | 0.6 | | 0.8 | |
| | AP | AUC | AP | AUC | AP | AUC | AP | AUC | AP | AUC | AP | AUC | AP | AUC | AP | AUC | AP | AUC |
| TGN | 63.89 | 59.97 | 65.15 | 61.07 | 65.58 | 61.46 | 54.66 | 53.12 | 55.67 | 53.99 | 56.57 | 54.66 | 66.63 | 59.41 | 67.46 | 60.33 | 67.72 | 60.88 |
| TGAT | 69.00 | 63.27 | 70.85 | 65.14 | 71.69 | 65.90 | 60.83 | 56.71 | 62.86 | 58.56 | 64.15 | 59.74 | 70.83 | 71.03 | 70.92 | 72.55 | 71.15 | 73.32 |
| GM_ori | 99.52 | 99.66 | 99.55 | 99.67 | 99.56 | 99.68 | 92.25 | 94.89 | 92.01 | 94.73 | 91.92 | 94.64 | 96.07 | 96.43 | 96.17 | 96.54 | 96.20 | 96.57 |
| GM_50n | 99.62 | 99.71 | 99.63 | 99.71 | 99.63 | 99.72 | 90.80 | 94.42 | 90.87 | 94.44 | 90.90 | 94.47 | 85.68 | 91.18 | 87.03 | 91.92 | 87.61 | 92.22 |
| DIDA | 64.16 | 63.16 | 66.08 | 64.71 | 67.35 | 65.67 | 53.33 | 54.86 | 53.34 | 54.96 | 53.24 | 54.29 | 64.25 | 66.50 | 65.66 | 68.15 | 66.59 | 69.26 |
| SIG | 99.85 | 99.90 | 99.86 | 99.89 | 99.90 | 99.92 | 99.88 | 99.93 | 99.94 | 99.97 | 99.92 | 99.96 | 99.85 | 99.86 | 99.79 | 99.81 | 99.81 | 99.84 |

## 6.4 EVALUATION ON OOD DATASETS (RQ3)

Following (Wu et al., 2022a), we generate the OOD datasets by injecting synthetic biases into the original dataset. For each node, we introduce two times the number of its existing connections as intervention edges. We employ three scales of 0.4, 0.6, and 0.8 to distinguish between positive and negative samples within the added intervention edges. Positive samples are drawn from the edges directly connected to the node, while negative samples are drawn from edges not connected to the node. Empirical results on OOD Datasets (Table 4) reveal the following observations: 1) SIG demonstrates superior performance across all datasets and distribution shift scales compared to existing baselines. While the best baseline, GM_ori, achieves comparable results to SIG on the IID datasets of Wikipedia, Reddit, and MOOC (Table 1), its performance drastically drops on OOD datasets. 2) SIG exhibits remarkable resilience to varying levels of distribution shift, indicating its ability to exploit invariant patterns under distribution shift scenarios. This robustness is particularly evident in the LastFM dataset, where SIG outperforms the best-performing baseline by nearly 8.00% in terms of AP.

## 7 CONCLUSION

This paper investigates a novel research problem, which simultaneously produces prediction results and explanatory subgraphs for continuous-time dynamic graphs. By analyzing this problem from a causal effect perspective, we introduce the ICCM, a novel causal inference model meticulously designed to address both IID and OOD scenarios for CTDGs. Building upon the theoretical foundations of ICCM, we propose a novel deep learning architecture, which translates theoretically established causal models into a practical solution for dynamic graphs. Our extensive empirical evaluations demonstrate the superior effectiveness and efficiency of the proposed SIG model, exhibiting significant advancements over existing methods in link prediction, explainability, and robustness when handling OOD data.

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

## A   NOTATIONS

## B   PRELIMINARIES

In this section, we introduce some necessary causal inference model concepts (Pearl et al., 2016) used in the paper.

Table 5: Notations and descriptions.

| Notations | Descriptions |
|---|---|
| $G = (\mathcal{V}, \mathcal{E}, \mathcal{T})$ | Dynamic graph $G$ with nodes set $\mathcal{V}$, edges set $\mathcal{E}$, and time domain $\mathcal{T}$ |
| $e_{ij}(t_k)$ | The edge between nodes $u_i$ and $v_j$ occurred at time $t_k$ |
| $x_{ij}^e(t_k)$ | The feature vector of $e_{ij}(t_k)$ |
| $x_i^n$ | The feature vector of $v_i$ |
| $G_s, G_t, G_b$ | Structural causal, temporal causal, and non-causal subgraph in causal model |
| $C_s, C_t$ | Constant structural and temporal subgraph in causal model |
| $M^*, H^*$ | Hidden representations in causal model |
| $Y$ | The prediction label in causal model |
| $U$ | Unobserved variables in causal model |
| $W_*^*, W_*^*$ | The model parameters |
| $\mathcal{D}$ | Confounder dictionary |
| $f_s, f_t$ | Structural and temporal encoding functions |
| $f_y^*(\cdot)$ | Linear networks |

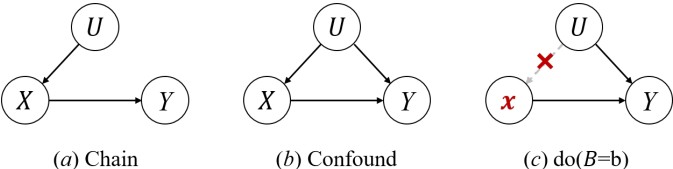

(a) Chain      (b) Confound      (c) do(B=b)

Figure 3: Basic causal inference models.

**Causal inference models**. The causal relationships between variables can be reflected by causal models. Figure 3 shows three instances of causal inference models. Chain (Fig. 3 (a)) represents sequential relationships where one variable influences another, which subsequently influences a third, and so on. Fig. 3 (b) illustrates the instance of confound. A variable $U$ is a confounder of the effect of $X$ on $Y$ if $U$ meets 3 conditions: $U$ is associated with $X$; $U$ is associated with $Y$ conditional on $X$; $U$ is not on a causal pathway from $X$ to $Y$. The confounder $U$ and backdoor path $X \leftarrow U \rightarrow Y$ make $Y$ and $X$ spuriously correlated. For instance, low blood pressure is seemingly linked to a higher risk of mortality. However, this association may be misleading, as it would be influenced by the confounding effect of heart disease. In this scenario, blood pressure ($X$) might appear as a direct cause of mortality ($Y$). Yet, the confounder heart disease ($U$) is associated with both low blood pressure and mortality.

**Do-operation.** The $do(X = x)$ operator is a mathematical tool used to simulate interventions within a model. As shown in Fig. 3 (c), it works by altering specific functions associated with $X$ in the model, replacing them with a constant $X = x$, while keeping the remaining model unchanged.

In the case of blood pressure ($X$) and mortality ($Y$), employing the $do(X = \text{'low'})$ or $do(X = \text{'normal'})$ operator entails fixing the blood pressure variable to a low or normal state for individuals. This intentional manipulation facilitates the analysis of mortality, particularly concerning the alteration in blood pressure, while holding other influential factors constant. Since it's impractical to collect data directly using the do-operation, adjustment formulas are proposed to compute the probability $P(Y = y \mid do(X = x))$. The adjustment formula is shown as follows:

$$
\begin{aligned}
& P(Y = y \mid do(X = x)) \\
&= \sum_{u \in \mathcal{D}} [P(Y = y \mid U = u, do(X = x))P(U = u \mid do(X = x))] \\
&= \sum_{u \in \mathcal{D}} [P(Y = y \mid U = u, X = x)P(U = u)].
\end{aligned}
\tag{17}
$$

Here, $\mathcal{D}$ represents the confounders dictionary, $P(Y = y \mid U = u, X = x)$ represents the probability considering the causal feature $X$ and confounding factors $U$, and $P(U = u)$ denotes the prior probability of these confounding factors. Note that $P(Y = y \mid X = x) \neq P(Y = y \mid do(X = x))$ unless there are no confounders present.

## C   MORE DETAILS ON SECTION 4 AND SECTION 5

### C.1   DETAILED EXPLANATIONS OF EQUATION 6 AND EQUATION 7

Equation (6) can be derived as follows:

$$
\begin{aligned}
&P(Y^S = y^S \mid do(G_s = C_s)) \\
&= \sum_{u \in \mathcal{D}} P(Y^S = y^S \mid U = u, do(G_s = C_s)) P(U = u \mid do(G_s = C_s)) \\
&= \sum_{u \in \mathcal{D}} P(Y^S = y^S \mid U = u, G_s = C_s) P(U = u) \\
&= \mathbb{E}_{u \in \mathcal{D}} \left[ P(Y^S = y^S \mid U = u, G_s = C_s) \right] \\
&= \mathbb{E}_{u \in \mathcal{D}} \left[ f_o^S(f_s^C(C_s), d) \right] \\
&= \mathbb{E}_{u \in \mathcal{D}} \left[ f_o^S(H^S, d) \right] \\
&= \mathbb{E}_{d \sim \mathcal{D}} [\sigma(W_1^c f_y^s(H^S) + W_3^c f_y^u(d))].
\end{aligned}
\tag{18}
$$

Similarly, we can derive Formula (7) as follows:

$$
\begin{aligned}
&P(Y^T = y^T \mid do(G_t = C_t)) \\
&= \sum_{u \in \mathcal{D}} P(Y^T = y^T \mid U = u, do(G_t = C_t)) P(U = u \mid do(G_t = C_t)) \\
&= \mathbb{E}_{d \sim \mathcal{D}} [\sigma(W_2^c f_y^t(H^T) + W_4^c f_y^u(d))].
\end{aligned}
\tag{19}
$$

### C.2   RISK FUNCTIONS

The risk functions are formulated as:

$$
\begin{aligned}
\mathcal{R}_i(y^I, y) &= \frac{1}{|\mathcal{E}|} \sum_{e \in \mathcal{E}} y_e log(y_e^I) + (1 - y_e) log(1 - y_e^I) \\
\mathcal{R}_t(y^T, y) &= \frac{1}{|\mathcal{E}|} \sum_{e \in \mathcal{E}} y_e log(y_e^T) + (1 - y_e) log(1 - y_e^T) \\
\mathcal{R}_s(y^S, y) &= \frac{1}{|\mathcal{E}|} \sum_{e \in \mathcal{E}} y_e log(y_e^S) + (1 - y_e) log(1 - y_e^S).
\end{aligned}
\tag{20}
$$

Here, the set $\mathcal{E}$ refers to the training dataset, which comprises pairs of positive and negative samples. Positive samples originate from the original edge sets, while negative samples are generated by substituting the destination nodes with randomly sampled nodes from the vocabulary, maintaining an equal ratio to the positive samples. The variable $y_e$ denotes the ground-truth label of edge $e$, assuming a value of 1 for positive samples and 0 for negative samples.

### C.3   TIME COMPLEXITY ANALYSIS

It takes $O(|P_1| \cdot N)$ time to extract the temporal causal graph and generate the temporal representation, where $|P_1|$ is the number of learnable parameters in the causal graph extractor $f_{ext}$, and $N$ is a hyperparameter denoting the number of recent edges for representation generation. It takes $O(|V_n| \cdot |x|)$ time to extract the structural causal graph and generate the structural representation, where $|V_n|$ is the number of nodes in the n-hop neighborhoods of two nodes for prediction and $|x|$ is the number of node features. The prediction takes $O(|P_2|)$ time, where $|P_2|$ is the number of parameters in the prediction model. Therefore, the total complexity of SIG is $O(|P_1| \cdot N + |V_n| \cdot |x| + |P_2|)$.

## D   FURTHER ANALYSES

### D.1   PROBLEM ANALYSIS

In the domain of explainable dynamic graph link prediction, a causal subgraph is extracted from the dynamic graph and elucidates the rationale behind the predicted label. A straightforward approach

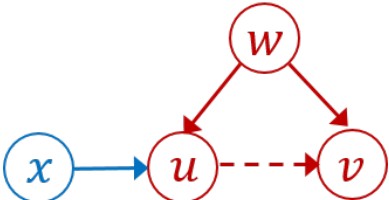

Figure 4: Example of shortcut features.

involves utilizing subgraph extraction techniques to extract the causal subgraph, from the initial graph $G$. Subsequently, link prediction is performed based on the information encoded within the extracted causal graph. Though this straightforward method may perform well w.r.t. IID data, its performance would downgrade when handling OOD data, as it is susceptible to the influence of confounding factors, i.e., variables correlated with both the causal subgraph and the target variable. These confounding factors can originate from the remaining subgraph of $G$ that is not encompassed by causal subgraph or can arise from latent and unobserved variables.

In the example in Figure 4, where a node $x$ consistently establishes a connection with node $u$ in the triadic closure pattern (red) within the training data. Though the triadic closure pattern is the reason for the link between $u$ and $v$, this straightforward method may tend to capture the bridging link (blue) rather than recognizing the specific triadic closure pattern. This bridging link could be a shortcut feature. In the test data, if the triadic closure pattern does not appear, the aforementioned models may still predict the link $(u, v)$ as long as it sees the bridging link. The presence of shortcut features makes it difficult to capture essential mechanisms, leading to inaccurate predictions. Therefore, it is crucial to carefully consider the potential for confounding factors when designing the model.

### D.2 Relationships with Related Models

This section delves into the connections between the proposed SIG framework and other relevant models in the research field.

**DIR** (Wu et al., 2022b) is an invariant rational discovery method specifically designed for static graphs. Similar to SIG, DIR mitigates spurious correlations between $G_c$ and $Y$ through the adoption of a do-operation. However, DIR's approach to performing the do-operation differs from that of SIG. DIR modifies non-causal elements in the graph to generate counterfactual graph data, while SIG employs the Normalized Weighted Geometric Mean (NWGM) approximation to efficiently estimate the causal effect without directly modifying the graph structure.

DIR's learning strategy is formulated as follows:

$$\min \mathbb{E}_c[\mathcal{R}(\hat{Y}, Y)|do(G_s = s)] + \lambda Var_s\big(\mathcal{R}(\hat{Y}, Y)|do(G_s = s)\big), \qquad (21)$$

where $\mathcal{R}$ represents the risk function, $\hat{Y}$ denotes the predicted label, and $\lambda$ controls the trade-off between minimizing interventional risks and their variance. DIR aims to minimize both interventional risks and their variance, ensuring that the model is not overly sensitive to specific interventions. However, these interventions can pose computational challenges, especially as graph sizes increase.

Without classifiers $f_o^I$ and $f_o^T$, and using the same do-operation implementation method as DIR, the proposed SIG methodology becomes equivalent to DIR. This demonstrates that SIG encompasses DIR as a special case, while capturing the temporal information and offering improved computational efficiency through the NWGM approximation.

**DIDA** (Zhang et al., 2022b) is an invariant rational discovery method specifically designed for DT-DGs. Its learning strategy aligns with that of DIR, aiming to minimize interventional risks and their variance. However, DIDA proposes an approximation to the intervention process by sampling and replacing the variant pattern representation instead of directly modifying the original graph structure. This approach aims to reduce the computational burden of interventions in DTDGs.

The probability function associated with DIDA's intervention process is expressed as follows:

$$P(Y = y|do(G_c = C)) = \mathbb{E}_{d \sim \mathcal{D}}[\text{Softmax}\big(g(z_c + z_d)\big)] \qquad (22)$$

Table 6: Summary of dataset statistics.

| Dataset | Wikipedia | Reddit | MOOC | LastFM | SX |
|---|---|---|---|---|---|
| #edge | 157,474 | 672,447 | 411,749 | 1,293,103 | 1,443,339 |
| #node | 8,227 | 10,000 | 7,047 | 1,980 | 194,085 |
| #dim-E | 172 | 172 | / | / | / |
| #dim-N | / | / | / | / | / |

where $z_c$ and $z_d$ represent the hidden representations for the cause and bias graph, respectively. $g(\cdot)$ makes predictions using both $z_c$ and $z_d$. Notably, DIDA requires the construction of an intervention set, denoted as $\mathcal{D}$, for each node and time step, which requires expensive sampling.

By omitting the classifier $f_o^I$, integrating structural and temporal interventions, discarding the NWGM approximation, and utilizing the same implementation approach, the proposed SIG methodology becomes equivalent to DIDA. This again highlights the generality of SIG and its ability to incorporate existing methods as special cases.

**GraphMixer (GM)** (Cong et al., 2023) presents a neural network architecture specifically designed for temporal graphs. Its main goal is to learn effective representations of temporal graphs for predictive tasks. If the causal subgraph extraction and the do-operation are omitted from the SIG framework, SIG reduces to GM.

In summary, SIG represents the first self-interpretable GNN tailored explicitly for both IID and OOD CTDGs. Temporal graph neural networks designed for CTDGs, such as GM, fail to provide explainable outcomes. Moreover, existing self-interpretable graph neural networks intended for static graphs (e.g., DIR) and DTDGs (e.g., DIDA) encounter limitations in their adaptation to CTDGs due to computational complexities. SIG effectively tackles these challenges by introducing two novel causal models, ICM and CCM. These meticulously designed models capture both temporal and structural information within CTDGs, simultaneously addressing confounding effects. Additionally, SIG specifies the essential components for implementing the causal models, including an extractor for identifying invariant subgraphs, two encoders for transforming subgraphs into latent representations, and classifiers for predictive modeling based on the derived causal graphs.

# E  EXPERIMENTAL SETTINGS

All the experiments are conducted on a computer with Intel(R) Core(TM)2 Duo CPU @2.40 GHz processor, 128 GB RAM, and Tesla T4.

## E.1  DATASETS.

We conducted experiments on five real-world datasets. The details of the datasets are reported in Table 6, where *#edge* and *#node* represent the number of node and edges. *#dim-E* and *#dim-N* denote the dimensions of node and edge features, respectively.

`Wikipedia`[1] captures edits made by Wikipedia editors over a month, with extracted link features derived by converting edit text into LIWC feature vectors (Pennebaker et al., 2001). `Reddit`[2] compiles posts from various subreddits within a month. The source node represents a user, while the target node denotes a subreddit. Each edge signifies a user's post in a specific subreddit. Similar to the Wikipedia dataset, link features are extracted through the conversion of text into LIWC feature vectors. `MOOC`[3] constitutes a bipartite network involving online resources. It comprises two kinds of nodes: students and units of course content. The connection between nodes signifies a student's interaction with specific content units. `LastFM`[4] serves as a commonly used dataset for music recommendation and analysis. It contains user listening histories and music tag information from

---

[1]http://snap.stanford.edu/jodie/wikipedia.csv

[2]http://snap.stanford.edu/jodie/reddit.csv

[3]http://snap.stanford.edu/jodie/mooc.csv

[4]http://snap.stanford.edu/jodie/lastfm.csv

the LastFM music platform. $\text{SX}^5$ stands as a temporal network of interactions on the Stack Exchange website "super user".

Note that for the datasets without node features, we utilize one-hot vectors as the nodes' features. Considering the large size of $\text{SX}$, it is impractical to use this manner. Hence, we randomly select 100 nodes for each node, and the corresponding shortest distances between them are used as the node's feature.

## E.2 Evaluation Metrics.

We partitioned the datasets based on the edge occurrence time: the initial 70% of edges were designated as the training set, the subsequent 15% were allocated to the validation set, and the remaining 15% formed the test set. We employ the average precision (AP) and area under the curve (AUC) as the evaluation metrics for link prediction. AP and AUC are two common metrics used to evaluate the performance of binary classification models. AP is a measure of the average precision across all possible recall thresholds. AUC is a measure of the area under the receiver operating characteristic (ROC) curve.

We adopt fidelity w.r.t. sparsity as the evaluation metrics for graph explanation. The definitions of Fidelity and sparsity are shown as follows:

$$Fidelity^{ap} = \frac{1}{N} \sum_{i=1}^{N} \big( ap(G) - ap(G_b) \big). \tag{23}$$

$$Sparsity = \frac{|G_c|_e}{|G|_e}. \tag{24}$$

Here, $N$ is the number of test graphs, $G$ represents the input graph, $G_b$ represents the residual portion of graph $G$ after excluding the explanatory subgraph $G_c$, $ap(G)$ represents the average precision output by graph $G$. $|G|_e$ ($|G_c|_e$) denotes the number of edges in $G$ ($G_c$). Higher values of $Fidelity^{ap}$ signify better explanatory outcomes, indicating the identification of more distinctive features. Lower values for sparsity indicate that the explanations are sparser and can focus primarily on more essential input information. Furthermore, we obtain the fidelity-sparsity curve and calculate the area under the curve (AUFSC) to evaluate interpretability performance, where a higher AUFSC value indicates better performance.

## E.3 Training Protocols.

An early-stopping mechanism was employed, terminating training when the Average Precision (AP) metric showed no improvement for five consecutive epochs. The model underwent training for 300 epochs using the Adam optimizer with a learning rate set at 0.0001 and a weight decay of 1e-6. We set the batch size to 600, and the hidden layer dimension to 100. For the extraction of the causal subgraph, we specified the number of recent edges ($N$) as 50 and employed 1-hop neighbors. All MLP layers were configured to 2. Regarding the link prediction task, negative samples were set at a ratio of 1:5 in the training set and adjusted to 1:1 in both the validation and test sets. Hyperparameters $\lambda_i, \lambda_t$, and $\lambda_s$ were set to 1.0, 0.5, and 0.5, respectively.

## E.4 Baselines

Note that the proposed SIG is the first self-interpretable GNN specifically designed for CTDGs. Given the limited studies in self-interpretable GNNs for dynamic graphs, our evaluation spans several comparisons by considering different types of baselines. (1) Initially, SIG undergoes comparison with three existing dynamic GNN models: TGN (Rossi et al., 2020), TGAT (Xu et al., 2020), and GraphMixer (GM) (Cong et al., 2023). These models are designed to handle CTDGs. However, as they lack the capacity to produce explainable outcomes, our comparison primarily focuses on link prediction tasks across original datasets and synthetic OOD datasets. (2) Additionally, we compare the proposed model with four post-interpretable models, including an attention-based explainer

---

[5] https://snap.stanford.edu/data/sx-superuser.html

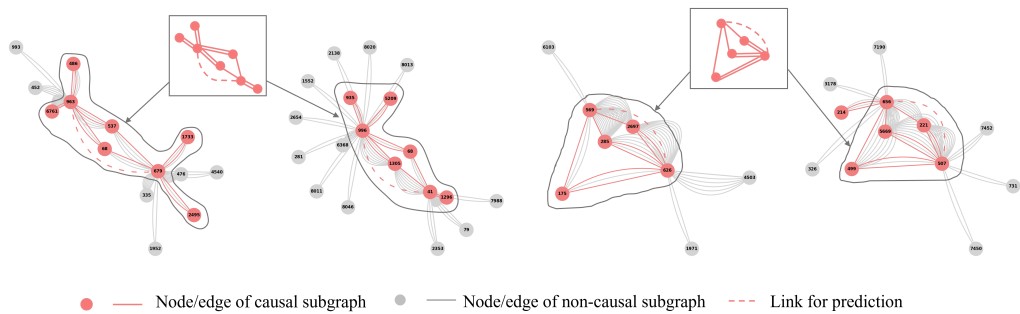

Figure 5: Case study of temporal casual subgraph ($G_t$) on the wikipedia dataset.

(ATTN (Xia et al., 2022)), a perturbing-based explainer (PBONE (Xia et al., 2022)), a static graph explainer (PGExp (Luo et al., 2020)), and a dynamic graph explainer (TGExp (Xia et al., 2022)). These models were thoughtfully chosen to represent diverse graph explanation approaches. Given their post-interpretable nature, our comparison focuses solely on graph explanation tasks. (3) Further, we compare SIG with DIDA (Zhang et al., 2022b), a self-interpretable GNN for DTDG. Our comparative analysis with DIDA spans across all tasks.

- **TGN** initially captures temporal information using Recurrent Neural Networks, followed by the graph attention convolution to jointly encompass spatial and temporal information.

- **TGAT** leverages a self-attention mechanism as its foundational element, incorporating a novel functional time encoding technique. TGAT can discern node embeddings as functions of time and can deduce embeddings for previously unseen nodes in an inductive manner.

- **GM** represents a straightforward architecture composed of three core components: a link-encoder utilizing MLPs, a node-encoder relying solely on neighbor mean-pooling, and an MLP-based link classifier. For **GM_ori**, we adhered to the default parameters provided in the paper's source code. Conversely, for **GM_50n**, we configured the number of recent edges to 50, aligning it with the setting used in SIG.

- **ATTN** extracts the attention weights in TGAT/TGN and averages the values over all layers. The averaged weights are regarded as importance scores.

- **PBONE** functions as a direct explainer by perturbing a single candidate edge. We configured the interpretation process for TGAT and TGN.

- **PGExp** employs a deep neural network to parameterize the generation process of explanations. In line with (Xia et al., 2022), we tailor it for temporal graph scenarios by computing weights for each event rather than each edge.

- **TGExp** comprises an explorer that identifies event subsets using Monte Carlo Tree Search and a navigator that learns event correlations to reduce the search space.

- **DIDA** represents self-interpretable GNN tailored explicitly for DTDGs. To enable a comparative analysis with the proposed model, we adjust our datasets by converting edges that occur within monthly periods into snapshots, thereby aligning our datasets with the DTDG setting.

## F SUPPLEMENTARY EXPERIMENTS

### F.1 CASE STUDY

Figure 5 depicts four examples of extracted temporal causal subgraphs ($G_t$), which are causal subgraphs based on temporal edges. This visualization demonstrates that the first two instances share similar causal graphs, as do the last two instances. Hence, the proposed SIG effectively extracts key topologies in the graph that are useful for dynamic link predictions.

Figure 6 depicts two examples of extracted structural causal subgraphs ($G_s$), which are based on structural node features. We observe that even for the same source node, different causal subgraphs are extracted when connected to different destination nodes. Note that the Wikipedia dataset contains abundant link features but no node features. Therefore, we use a one-hot vector as its node features.

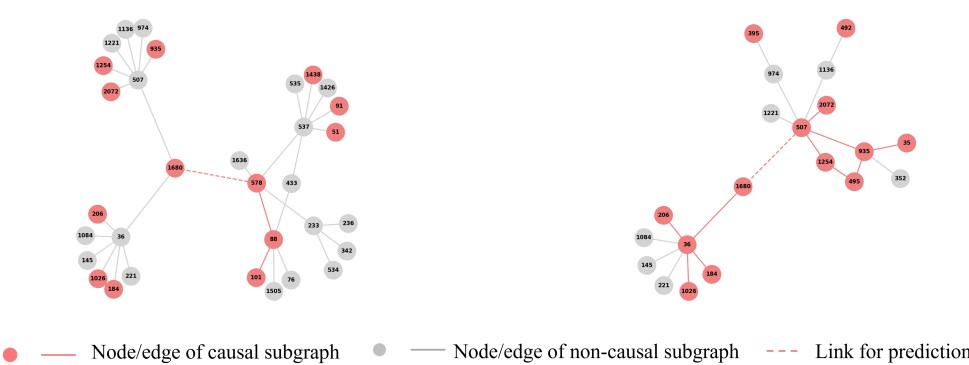

Node/edge of causal subgraph · Node/edge of non-causal subgraph --- Link for prediction

Figure 6: Case study of structural casual subgraph ($G_t$) on the wikipedia dataset.

## F.2 EFFICIENCY

Table 7 shows the efficiency of our method w.r.t all dynamic GNN baselines. TGAT and TGN exhibit slower performance compared to SIG due to their utilization of more complex encoding networks. Additionally, DIDA operates at a slower pace than SIG as it necessitates gathering a confounder dictionary in each snapshot. GM demonstrates slightly better efficiency than SIG, because GM does not output an explainable subgraph.

Table 7: Average running time per edge (seconds).

|  | Self-int | | Dynamic GNN | | | | Post-hoc | | | |
|---|---|---|---|---|---|---|---|---|---|---|
|  | SIG | DIDA | TGN | TGAT | GM_ori | GM_50n | ATTN | PBONE | PGExp | TGExp |
| Reddit | $5.8\times10^{-4}$ | $3.7\times10^{-3}$ | $1.7\times10^{-3}$ | $4.2\times10^{-3}$ | $2.2\times10^{-4}$ | $3.7\times10^{-4}$ | 0.95 | 1.10 | 0.82 | 412.65 |
| LastFM | $6.2\times10^{-4}$ | $2.8\times10^{-3}$ | $1.4\times10^{-3}$ | $3.7\times10^{-3}$ | $1.5\times10^{-4}$ | $2.1\times10^{-4}$ | 3.44 | 2.89 | 2.59 | 716.08 |

Table 7 also illustrates the efficiency comparison of our method against all graph explanation methods. Notably, all post-hoc explainable GNNs exhibit high computational costs, leading to delayed detections. Each of these methods requires over 0.8 seconds to explain an edge. Among the baselines, the self-interpretable GNN model, namely DIDA, emerges as the most efficient baseline. However, despite its efficiency, DIDA's speed remains significantly slower than SIG. This discrepancy arises from DIDA's necessity to gather a confounder dictionary in each snapshot, a process that is time-consuming.

## F.3 ABLATION STUDY

We conducted ablation studies by removing ICM, temporal, and structural classifiers. The ablation experiments are summarized in Table 8. The results reveal that the complete solution achieves the highest performance, validating the efficacy of our proposed design. Specifically, we notice that ICM significantly contributes to the performance in both the original and OOD datasets. Moreover, the removal of structural and temporal losses results in marginal performance changes in the original dataset, whereas their absence notably impacts the performance in OOD datasets, indicating their substantial contribution in handling out-of-distribution scenarios.

Table 8: Ablation study on various original and OOD graph datasets.

| Dataset | Reddit | | LastFM | | SX | | Reddit_OOD | | LastFM_OOD | | SX_OOD | |
|---|---|---|---|---|---|---|---|---|---|---|---|---|
|  | AP | AUC | AP | AUC | AP | AUC | AP | AUC | AP | AUC | AP | AUC |
| remove structural classifier | 99.88 | 99.89 | 98.37 | 99.03 | 99.16 | 99.23 | 74.22 | 71.73 | 50.15 | 49.97 | 93.01 | 95.07 |
| remove temporal classifier | 99.87 | 99.88 | 97.75 | 98.51 | 99.53 | 99.48 | 98.37 | 98.19 | 78.29 | 82.90 | 99.31 | 99.17 |
| remove ICM | 98.76 | 98.66 | 53.96 | 54.95 | 95.86 | 97.14 | 97.75 | 97.71 | 50.38 | 50.48 | 95.27 | 95.07 |
| SIG | **99.99** | **99.99** | **99.96** | **99.98** | **99.71** | **99.70** | **99.90** | **99.92** | **99.92** | **99.96** | **99.81** | **99.84** |

Figure 7 shows the throughput of our solution by varying the number of edges $N$ and the number of hops $n$. We observe that the change of throughput is linear to the hyperparameter $N$, which is

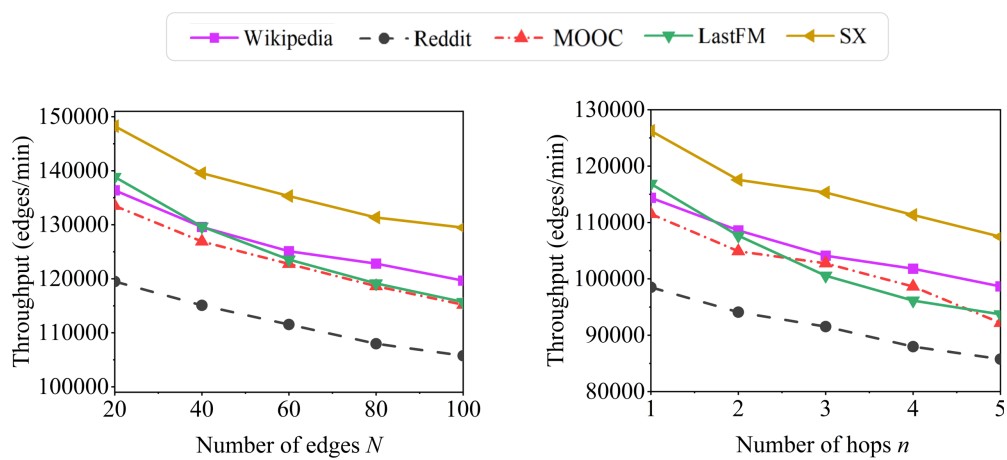

Figure 7: Throughput w.r.t. hyper-parameters on five graph datasets.

consistent with the complexity analysis. As the number of hops $n$ increases, the number of nodes in the extracted structural causal graph increases greatly, reducing the throughput in a linear trend.