# OpenReview forum: "SIG: Self-Interpretable Graph Neural Network for Continuous-time Dynamic Graphs"
_ICLR.cc/2025/Conference — Submitted to ICLR 2025_

### Official Review · Reviewer_iLVa · 2024-10-30

**Soundness:** 2
**Presentation:** 2
**Contribution:** 2
**Rating:** 5
**Confidence:** 2

**Summary:**

The paper studies the problem of self-interpretable GNNs for continuous-time dynamic graphs (CTDGs), i.e., predict labels while generating a causal subgraph for interpretation. The authors first provide a theoretical analysis of the causal effect, which motivates the proposed causal inference model, the Independent and Confounded Causal Model (ICCM). Based on ICCM, the paper proposes self-interpretable GNN (SIG) to predict labels while offering an explanation. Finally, the paper conducts extensive experiments to demonstrate the effectiveness of the proposed method.

**Strengths:**

- The paper studies a valuable problem about self-interpretable GNNs for CTDGs.

- The authors conduct thorough experiments to verify the performance of link prediction and explanation capability for SIG.

**Weaknesses:**

- The paper writing needs to be improved for clarity, e.g.,

&emsp; (1) The goal of the paper is to conduct link prediction while offering an explanation for prediction, the proposed method should be SIG, but it only introduces ICCM for causal analysis in the Abstract.

&emsp; (2) The relationship between SIG and ICCM confuses readers before they understand Sections 4 and 5.

&emsp; (3) A real-world case for confounders in CTDGs should be offered to better understand causal analysis and the design of SIG.

- Confounder mining is done by clustering the representations of edges, but there is a lack of explanation on how this process works.

- In Section 5.4, the proposed method is optimized by maximizing mutual information. Is the goal of this optimization to enable the model to capture causal subgraphs? If so, would maximizing the mutual information between the predicted relevant subgraphs and labels also make the model capture spurious connections, thus having the opposite effect?

- Similar performance can be observed in DIDA (Table 1) and SIG without ICM (Table 8). A deeper analysis is needed to explain this phenomenon, focusing on the similarities between DIDA and SIG without ICM.

**Questions:**

See above.

---

> ### Author Response · Authors · 2024-11-23
>
> ### W1
>
> **Comment:**
> Writing problem: The relationship between **SIG** and **ICCM** should be introduced in the Abstract. A real-world case for confounders in CTDGs should be offered.
>
> **Response:**
> Thank you for your valuable feedback. **SIG** is the proposed method, which is built upon the theoretical foundation of causal inference provided by **ICCM** and incorporates a deep learning application designed based on ICCM. We will update the Abstract to explicitly clarify the relationship between **SIG** and **ICCM** in the revised version.
> Regarding the real-world case for confounders in CTDGs, we have introduced one in **Appendix D.1: Problem Analysis**. To enhance accessibility, we will move this content to the main paper in the updated version.
>
> ### W2
>
> **Comment:**
> Confounder mining is done by clustering the representations of edges, but there is a lack of explanation on how this process works.
>
> **Response:**
> Thank you for your comment. The confounder dictionary is designed to capture all confounders, including non-causal subgraphs (e.g., shortcut features) and unobserved factors. This dictionary enables us to incorporate both the causal subgraph and the confounders into the **do-operation**. However, directly combining causal subgraphs and confounders is computationally inefficient.
>
> To address this, instead of explicitly combining causal subgraphs with confounders, we operate on their representations. The temporal and structural subgraph representations of each link encode both the causal subgraph and the confounders relevant to that link. Notably, what constitutes a causal subgraph for one link might be a non-causal subgraph for another. This allows us to use the encoded representation to approximate potential confounders.
>
> Given the dynamic nature of the graph, the number of distinct confounders can grow extremely large, making it computationally expensive to account for all possible confounders. To mitigate this, we cluster the representations and use the centroids of these clusters to approximate sets of similar confounders. This clustering approach significantly improves efficiency while preserving the ability to capture the diverse range of confounders in the graph.
>
>
> ### W3
>
> **Comment:**
> Would maximizing the mutual information between the predicted relevant subgraphs and labels also make the model capture spurious connections, thus having the opposite effect?
>
> **Response:**
> Thank you for your question. Without the constraints **Cs⊥U** and **Ct⊥U**, maximizing the mutual information between the predicted relevant subgraphs and labels could indeed lead the model to capture spurious connections. In fact, without these constraints, the causal inference model reduces to the standard **ICM** framework, which does not account for confounding effects. This limitation is precisely why we extend **ICM** to **ICCM**, incorporating these constraints to ensure a more robust causal analysis.
>
> ### W4
>
> **Comment:**
> Similar performance can be observed in **DIDA** (Table 1) and **SIG** without ICM (Table 8). A deeper analysis is needed to explain this phenomenon, focusing on the similarities between DIDA and **SIG** without ICM.
>
> **Response:**
> Thank you for your insightful comment. **DIDA** is an invariant rational discovery method specifically designed for DTDGs. However, it requires the construction of an intervention set for each node and time step, which involves computationally expensive sampling.
>
> We have provided a detailed theoretical analysis of the relationship between **DIDA** and **SIG** in **Appendix D.2: Relationships with Related Models**. To address your concern and improve accessibility, we will incorporate this analysis into the main paper in the revised version.

---

> > ### Comment · Reviewer_iLVa · 2024-11-29
> >
> > Thank you for your reply. Your response has addressed most of my concerns, but it seems I have not seen the updated revised PDF. Additionally, it appears that the issues raised by other reviewers have not been resolved. So, I maintain my score.

---

### Official Review · Reviewer_bg3P · 2024-11-02

**Soundness:** 2
**Presentation:** 3
**Contribution:** 3
**Rating:** 5
**Confidence:** 4

**Summary:**

The paper introduces a novel task for CTDGs aimed at predicting labels and generating explanations through causal subgraphs. It presents multiple insights from a causal analysis of SIG, along with an implementation of the method. SIG effectively captures both structural and temporal aspects of the data. Extensive experimental results compare SIG with various methods in the literature, evaluating link-prediction accuracy and explainability. Overall, SIG outperforms the provided baselines in terms of both accuracy and efficiency.

**Strengths:**

1) The paper addresses critical fields: link prediction, dynamic graphs, and explainability, which hold long-term relevance for both academia and industry.
2) By sharing the code and providing a detailed description of how to run it, the authors enhance the paper's reliability.
3) The authors first present the insights and theoretical foundations of the method, followed by its practical implementation.

**Weaknesses:**

Although multiple experimental results are provided, there are certain flaws that need to be addressed.

1) Following the sentence, "Regarding the link prediction task, negative samples were set at a ratio of 1:5 in the training set and adjusted to 1:1 in both the validation and test sets" (Line 905), I understand that the test set has been sampled, which should not be the case. If sampling is applied, that will introduce biases into the results. I would expect the authors to conduct testing on the complete set of test edges and provide precision@k (where k can be the number of positive edges or its small multiplier) results to better understand the method's true quality in real-life setting.
2) To avoid bias, I won’t specify paper names, but I believe the method should be compared to studies from 2023 and 2024 at top ML/AI conferences for explainable AI methods for graphs and dynamic graph architectures.
3) The method uses multiple components from Cong et al., which already performs well on the provided link-prediction datasets. I recommend that the authors compare methods under unbiased testing conditions and, if possible, add a more challenging dataset where Cong et al. does not perform well.
4) The experiments need to be run multiple times to see the deviations between runs to test robustness, not just with a single random seed.

**Questions:**

My main concerns are around the experiment results. I liked the development of the paper and uses of the idea of causal inference model. I'm open to improve my score if the (some) items are provided. If not provided, I would like to hear the explanations from the authors. So to summarize my questions:

1) If the sampling is applied to the testing, can authors provide full-test results (i.e., without sampling test edges), and provide different metric, precision@k where k is the number of positive edges or its small multiplier?
2) Can authors provide more recent baselines for both tasks, link prediction and explainability?
3) Can authors provide more deeper analysis about the differences between Cong et al., and SIG on the experiment results? Can authors find more challenging dataset which can demonstrate SIG's capabilities beyond just building on Cong et al.'s strong performance?
4) Can authors run their method multiple times (at least 5 times with different seeds) and provide mean/standard deviation across runs?

---

> ### Author Response · Authors · 2024-11-23
>
> ### W1/Q1
>
> **Comment:**
> The test set has been sampled, which should not be the case.
>
> **Response:**
> Thank you for your comment. We would like to clarify that all edges in the test set are indeed used during our evaluation. In the link prediction task, the model is tasked with predicting "true" for edges in the test set (positive samples) and "false" for node pairs that are not linked by an edge (negative samples).
>
> When we stated, "negative samples were set at a ratio of 1:5 in the training set and adjusted to 1:1 in both the validation and test sets," we meant that we sampled a subset of unconnected node pairs as negative samples, rather than using all disconnected node pairs.
>
> It is important to note that using all disconnected node pairs as negative samples is computationally infeasible for dynamic graphs, particularly those with a large number of nodes and edges. Negative sampling is a standard practice in link prediction tasks to manage computational complexity effectively.
>
> For instance, the **SX** dataset contains 194,085 nodes and 1,443,339 edges. At each timestep, there is one positive pair and approximately **37.67 billion** potential negative pairs. Such computational demands are clearly impractical.
>
>
> ### W2/Q2
>
> **Comment:**
> Can authors provide more recent baselines for both tasks, link prediction and explainability?
>
> **Response:**
> Thank you for your valuable feedback. We have conducted a thorough evaluation against state-of-the-art methods presented at top ML/AI conferences, using eight baselines. We believe this selection provides a robust and comprehensive basis for assessing our approach.
>
> Furthermore, as stated in the paper, to the best of our knowledge, the proposed **SIG** framework is the first self-interpretable GNN for Continuous-Time Dynamic Graphs (CTDGs) capable of handling both IID and OOD data. If the reviewer is aware of any additional recent methods for comparison, we would be grateful for specific references and would gladly include them in future work.
>
>
> ### W3/Q3
>
> **Comment:**
> Cong et al. already performs well on the provided link-prediction datasets. Provide a deeper analysis of the differences between Cong et al. and SIG based on the experimental results.
>
> **Response:**
> Thank you for your comment. The datasets used in our comparisons are widely recognized benchmarks for link prediction tasks. While Cong et al.'s method demonstrates strong performance on most IID datasets (as shown in Table 1), our proposed **SID** framework significantly outperforms **GM_ori** on OOD datasets (as demonstrated in Table 5).
>
> These results highlight the key distinction between the two methods: **SID** is specifically designed to handle distribution shifts, providing robust performance in OOD scenarios, which is the primary focus of this paper. This capability makes **SID** particularly well-suited for real-world applications where test distributions often differ from training distributions. We believe this distinction underscores the novelty and value of our approach.
>
>
> ### W4/Q5
>
> **Comment:**
> The experiments need to be run multiple times to see the deviations between runs to test robustness.
>
> **Response:**
> Thank you for your comment. We would like to clarify that the reported results are indeed the average of five independent runs with different random seeds. We would clarify this in the amended version.

---

> > ### Comment · Reviewer_bg3P · 2024-11-24
> > **Follow up.**
> >
> > Thank you so much for your answers to my questions.
> >
> > Regarding Q1: Sampling negative edges add lots of biases and it is not acceptable in real life settings. For instance, if you want to decide which two nodes are needed to be connected for recommendation systems, you have to give scores to the all pairs. I understood that when there are billions of edges, single threaded applications will not be able to solve it in a reasonable time, however 1:1 still introduces too much biases. I would suggest authors to use 1:50, 1:100 ratios for those big datasets, and full test for smaller datasets.
> >
> > Regarding Q4: Please correct me if I’m wrong but I still cannot see the standard deviation between runs. Can you provide those in the revised version?

---

### Official Review · Reviewer_c7Kh · 2024-11-04

**Soundness:** 2
**Presentation:** 3
**Contribution:** 2
**Rating:** 5
**Confidence:** 3

**Summary:**

This paper studies the self-interpretable GNNs for continuous-time dynamic graphs. The authors propose a Independent and confounded causal model which incorporates independent causal model and confounded causal model. Experimental results demonstrate the proposed method works well on both IID and OOD data.

**Strengths:**

1. This papers study the model explanation on dynamic graphs, which consists of both temporal information and structure information.
2. The authors consider both IID and OOD setting.
3. The paper is well-written and easy to follow.

**Weaknesses:**

1. Although the authors claim to provide theoretical analysis, I found no theoretical insight specifically focused on the proposed method.

2. The datasets chosen appear relatively simple, as several baseline methods achieve close to 100% AUC and AP, limiting the scope of meaningful comparison.

3. The baselines included are limited and somewhat outdated. More recent relevant works, such as [1, 2, 3], should be considered to provide a more comprehensive evaluation.

4. For the temporal causal subgraph extraction, the authors only extract most recent temporal edges, neglecting the graph's evolution over time and potentially overlooking key historical patterns that inform causal relationships.

[1] Wang, Tianchun, et al. "DyExplainer: Explainable Dynamic Graph Neural Networks." arXiv preprint arXiv:2310.16375 (2023).
[2] Wang, Jie, et al. "Trustguard: Gnn-based robust and explainable trust evaluation with dynamicity support." IEEE Transactions on Dependable and Secure Computing (2024).
[3] Liang, Guojun, et al. "Dynamic Causal Explanation Based Diffusion-Variational Graph Neural Network for Spatiotemporal Forecasting." IEEE Transactions on Neural Networks and Learning Systems (2024).

**Questions:**

1. What is the difference between discrete-time dynamic graphs  and continuous time dynamic graphs? The  continuous time  can be divided into several discrete steps.

2. How to define the ground truth for an explanation subgraph?

---

> ### Author Response · Authors · 2024-11-22
> **Response to Reviewer c7Kh (Part 1)**
>
> **Q1:**  What is the difference between discrete-time dynamic graphs and continuous time dynamic graphs? The continuous time can be divided into several discrete steps.
>
> **Answer** : Discrete-time dynamic graphs (DTDGs) are represented as a sequence of snapshots, with each snapshot capturing the state of the graph at a specific time step. In contrast, continuous-time dynamic graphs (CTDGs) are represented as a sequence of events, each associated with a precise timestamp.
>
> While continuous time can be divided into discrete steps, replacing CTDGs with DTDGs may result in the loss of fine-grained temporal information, such as the exact timing and order of events.
>
> **Q2:** How to define the ground truth for an explanation subgraph?
>
> **Answer:** Since it is not possible to define the ground truth for an explanation subgraph in a CTDG, we adopt "fidelity," a widely used quantitative metric for graph explanation, to evaluate our method in the experiments.
>
> **W1:**  Although the authors claim to provide theoretical analysis, I found no theoretical insight specifically focused on the proposed method.
>
> **Response:** We appreciate the reviewer’s comment and the opportunity to clarify. Our proposed deep learning model is indeed based on a solid theoretical foundation, specifically addressing the challenge of efficiency when applying causal inference theory to dynamic graph neural networks. The model itself is a direct outcome of this theoretical analysis, which underpins its development.
>
> We acknowledge that the theoretical aspects may not be immediately clear on first reading. However, we believe the framework is deeply rooted in causal inference principles, particularly in their application to continuous dynamic graph neural networks. As such, the claim that the method lacks theoretical insight is not accurate.
>
> **W2:** The datasets chosen appear relatively simple, as several baseline methods achieve close to 100% AUC and AP, limiting the scope of meaningful comparison.
>
> **Response:**  Thank you for your feedback. The datasets used in our study are widely recognized in the literature for evaluating dynamic graph neural networks. We selected these datasets based on their relevance and prior use in similar studies, which is why their inclusion is not problematic. However, we would be happy to consider additional datasets if the reviewer can suggest any.
>
> Furthermore, we would like to clarify that the focus of our paper is on addressing the challenges of out-of-distribution (OOD) detection and providing explainability. The near 100% AUC and AP scores observed with baseline methods are typical for standard datasets, but they do not reflect the more complex scenarios involving OOD datasets, which are central to our experiments. Therefore, the simplicity of these datasets should not diminish their relevance in the context of OOD detection and the explanatory aspects of our work.

---

> ### Author Response · Authors · 2024-11-22
> **Response to Reviewer c7Kh (Part 2)**
>
> **W3:** The baselines included are limited and somewhat outdated. More recent relevant works, such as [1, 2, 3], should be considered to provide a more comprehensive evaluation.
>
> **Response:**  Thank you for your comments. We would like to clarify that [1] is an arXiv preprint and has not been formally published, so it is not essential for comparison. Additionally, DyExplainer [1], Trustguard [2], and DVGNN [3] are designed for Discrete-Time Dynamic Graphs (DTDG), while our work focuses on Continuous-Time Dynamic Graphs (CTDG). Our paper have demonstrated that DTDG explanation methods are not suitable for CTDG due to efficiency constraints. This is also reflected in the datasets used in these papers, which are significantly smaller than those in our study. For example, Trustguard is evaluated on datasets with 32,029 and 22,650 edges, while DVGNN is tested on the PeMS08 traffic prediction dataset, which includes 170 detectors on 8 roads.
>
> [1] Wang, Tianchun, et al. "DyExplainer: Explainable Dynamic Graph Neural Networks." arXiv preprint arXiv:2310.16375 (2023).
> [2] Wang, Jie, et al. "Trustguard: Gnn-based robust and explainable trust evaluation with dynamicity support." IEEE Transactions on Dependable and Secure Computing (2024). [3] Liang, Guojun, et al. "Dynamic Causal Explanation Based Diffusion-Variational Graph Neural Network for Spatiotemporal Forecasting." IEEE Transactions on Neural Networks and Learning Systems (2024).
>
>
> **W4:** For the temporal causal subgraph extraction, the authors only extract most recent temporal edges, neglecting the graph's evolution over time and potentially overlooking key historical patterns that inform causal relationships.
>
> **Response:**  Thank you for your comment. In the context of Continuous-Time Dynamic Graphs (CTDGs), the sheer volume of data makes it impractical to analyze the entire graph. By focusing on recent temporal edges, we aim to strike a balance between capturing relevant causal relationships and maintaining computational feasibility. This approach is not intended to neglect the historical evolution of the graph, but rather to offer a practical solution that is both efficient and effective for analyzing large-scale temporal data.
>
> Furthermore, when the number of recent temporal edges is defined to match the total number of edges in the graph, our method effectively extracts all edges. Our experiments have shown that this approach successfully captures key historical patterns and causal relationships without the need to process the entire graph.

---

> > ### Comment · Reviewer_c7Kh · 2024-11-23
> >
> > Thanks for the authors' response. I still have some concerns:
> >
> > W1: In the introduction, the authors claim
> > > SIG initiates with a theoretical analysis of the problem from a causal effect perspective and proposes a novel causal inference model, namely the Independent and Confounded Causal
> > Model (ICCM).
> >
> > I still didn't understand where the theoretical analysis is.
> >
> > W2: If the performance is 100% AUC and AP, then where is the OOD?
> >
> > W3: I find other reviewers also have the same question about baselines, but there are still no recent baselines.

---

> > > ### Author Response · Authors · 2024-11-24
> > >
> > > Thank you for your follow-up response. I will address this question in more detail.
> > >
> > > **Question:** Where is the theoretical analysis?
> > > **Answer:** The theoretical analysis is presented in **Section 4: Causal Effect Look**.
> > >
> > > Our analysis focuses on extending the existing "do-operation" and adjustment formulas from causal inference models (see **Appendix B: Preliminaries**) to the context of link prediction in continuous-time dynamic graphs (CTDGs).
> > >
> > > The "do-operation" and adjustment formulas require interventions on the input. In dynamic graphs, the input is the causal subgraph, and the interventions correspond to all possible topologies. However, these adjustment formulas cannot be directly applied to link prediction in CTDGs. Unlike static or discrete-time graphs, CTDGs evolve continuously, with nodes and edges being added or removed over time. This results in a significantly larger space of possible topologies, greatly increasing the computational complexity of interventions in causal inference models (also discussed as the second challenge in the **Introduction**).
> > >
> > > To address this, we propose a novel causal inference model tailored for CTDGs, called **ICCM** (see Equations 8 and 9). ICCM extends the original adjustment function by introducing new functions that effectively eliminate the effects of confounders while remaining computationally efficient for dynamic graphs.
> > >
> > > **Section 4: Causal Effect Look** outlines the derivation process of ICCM, including how Equations 8 and 9 were obtained.
> > >
> > > **Question:** If the performance is 100% AUC and AP, then where is the OOD?
> > > **Answer:** We would like to clarify that the performance is not 100%.
> > >
> > > Our work evaluates two types of datasets:
> > > 1. **IID datasets** (results shown in Table 1).
> > > 2. **OOD datasets** (results presented in Table 4).
> > >
> > > Our proposed SID framework demonstrates significant improvements over GM_ori dataset specifically on OOD datasets.
> > >
> > >
> > > **Question:** : I find other reviewers also have the same question about baselines, but there are still no recent baselines.
> > > **Answer:**
> > > Thank you for your valuable feedback.
> > >
> > > We have already included eight baselines in our evaluation, five of which were proposed within the last three years. Furthermore, to the best of our knowledge, the proposed SIG framework is the first self-interpretable GNN for Continuous-Time Dynamic Graphs (CTDGs) capable of addressing both IID and OOD scenarios. If the reviewer is aware of any additional recent methods that would be suitable for comparison, we would greatly appreciate specific references and would be happy to consider them in the revision.
> > >
> > > In addition, we can incorporate more comparisons with methods from Discrete-Time Dynamic Graphs (DTDGs) in the revised version of the paper and conduct the necessary experiments. That said, we believe the current baselines are sufficient to demonstrate the superiority of the proposed method.

---

### Meta-Review · Area_Chair_HTQ9 · 2024-12-17

**Metareview:**

### Summary
The paper introduces Self-Interpretable GNN (SIG) for continuous-time dynamic graphs (CTDGs), addressing the task of link prediction while generating causal subgraphs for explanation. SIG is motivated by a proposed Independent and Confounded Causal Model (ICCM), which incorporates both independent and confounded causal effects. The model combines causal inference with GNN-based message passing, validated through experiments on synthetic and real-world datasets.

### Strengths
- The paper addresses self-interpretable GNNs for CTDGs, a relevant problem involving temporal and structural information, with applications in explainable dynamic graph models.
- The proposed ICCM introduces causal reasoning for extracting subgraphs that explain predictions, adding interpretability to GNN models.
- The paper evaluates the method's performance on link prediction and explainability, covering multiple datasets and settings (IID and OOD).

### Weaknesses
- While the paper claims to provide theoretical analysis, the causal effect analysis lacks depth, and there is no specific theoretical insight focused on the proposed method.
- The experimental evaluation excludes recent and relevant methods (e.g., DyExplainer, CAWN, DySAT), which weakens the comprehensiveness of comparisons. Results on simple datasets also limit the scope of meaningful evaluation.
- Methodological Ambiguity: The relationship between SIG and ICCM is confusing in the abstract and early sections.
Confounder extraction through clustering lacks clear explanation. Mutual information optimization (Section 5.4) raises concerns about capturing spurious connections instead of causal relationships.

The paper tackles a meaningful problem, but it suffers from unclear theoretical insights, limited experimental evaluation, and methodological ambiguity. The exclusion of recent baselines and reliance on simple datasets limit the significance of the results. Additionally, key aspects of the proposed causal inference process require clearer explanations and deeper analysis to validate the method's novelty and robustness.

**Additional Comments On Reviewer Discussion:**

The major concerns raised by the reviewers are:

- Lack of Theoretical Clarity: The claimed theoretical analysis lacks depth and specific insights focused on the proposed method.
Key components like confounder mining and the optimization objective (e.g., mutual information maximization) are unclear, raising concerns about potential spurious connections.
- Limited Experimental Evaluation: The evaluation excludes recent baselines (e.g., DyExplainer, CAWN, DySAT) from 2023–2024, reducing the comprehensiveness of comparisons. The experiments use simple datasets where baseline methods perform close to 100%, limiting the scope for meaningful conclusions. Moreover, standard deviations are not provided.
- Ambiguity in Methodology: The relationship between ICCM and SIG is not clearly explained, leading to confusion in the early sections.
The process of confounder mining through clustering lacks sufficient detail. Issues such as biased sampling in the test set and the absence of robust evaluations (e.g., multiple runs with deviations) undermine the reliability of the results.

Authors tried to address the above points, however, issues regarding recent baselines and standard deviation between runs are still not provided. Moreover, the setting of sampling test data is not realistic.

---

### Decision · Program_Chairs · 2025-01-22

Reject